# AUPO - ABSTRACTED UNTIL PROVEN OTHERWISE: A REWARD DISTRIBUTION BASED ABSTRACTION ALGORITHM

## ABSTRACT

We introduce a novel, drop-in modification to Monte Carlo Tree Search's (MCTS) decision policy that we call *AUPO*. Comparisons based on a range of IPPC benchmark problems show that AUPO clearly outperforms MCTS. AUPO is an automatic action abstraction algorithm that solely relies on reward distribution statistics acquired during the MCTS. Thus, unlike other automatic abstraction algorithms, AUPO requires neither access to transition probabilities nor does AUPO require a directed acyclic search graph to build its abstraction, allowing AUPO to detect symmetric actions that state-of-the-art frameworks like ASAP struggle with when the resulting symmetric states are far apart in state space. Furthermore, as AUPO only affects the decision policy, it is not mutually exclusive with other abstraction techniques that only affect the tree search.

## 1 INTRODUCTION

A plethora of important problems can be viewed as sequential decision-making tasks such as autonomous driving (Liu et al., 2021), energy grid optimization (Sogabe et al., 2018), financial portfolio management (Birge, 2007), or playing video games (Silver et al., 2016). Though arguably state-of-the-art on such decision-making tasks is achieved using machine learning (ML) as demonstrated by DeepMind with their AlphaGo agent for Go (Silver et al., 2016) or OpenAI Five for Dota 2 (Berner et al., 2019), there is still a demand for general domain-knowledge independent, on-the-go-applicable planning methods, properties which ML-based approaches usually lack but which are satisfied by Monte Carlo Tree Search (Browne et al., 2012) (MCTS), the method of interest for this paper. For example, Game Studios rarely implement ML agents as they have to be costly retrained whenever the game and its rules and updated. Though not within the scope of this paper, improvements to MCTS might also potentially translate to ML-based methods that use MCTS as their underlying search.

One family of approaches to improve the performance of MCTS is using abstractions that usually group similarly behaving nodes or actions of the search tree. State-of-the-art abstraction tree searches such as OGA-UCT (Anand et al., 2016) all rely on the reward function being deterministic, having full access to the transition probability of any sampled state-action pair, and on the search graph being a directed acyclic graph. While these methods could in principle still be applied if the first two conditions aren't met (i.e. by, approximating the reward and transition probabilities), they fundamentally rely on doing search on a DAG, which requires being able to check state equalities which is not always guaranteed (e.g., in memory-constrained settings where states may only be represented as action, stochastic-outcome sequences, in partially observable domains, in continuous-state settings, or in blackbox simulation settings). Until now, no domain-independent, non-learning-based MCTS abstraction algorithms for discrete, fully-observable settings exist that have no additional constraints than MCTS, exist. This is a gap that this paper closes.

Concretely, we introduce **A**bstracted **U**ntil **P**roven **O**therwise (AUPO), the first MCTS-based abstraction algorithm that can significantly outperform MCTS in a discrete, fully-observable, non-learning-based setting whilst requiring neither access to transition probabilities nor a directed acyclic search graph, nor a deterministic reward setting. AUPO only affects the decision policy and can thus even be combined and enhanced with other abstraction algorithms during the tree policy.

Furthermore, in practice, AUPO can detect symmetric actions that the ASAP (Anand et al., 2015) framework cannot when the resulting symmetric states are far apart in state space, as ASAP needs the search graph to converge. As only the decision policy is affected, AUPO's runtime overhead vanishes with an increase in the iteration count (see Tab. 3).

The key idea of AUPO is to consider all actions at the root node initially as equivalent, only separating them if the layerwise reward distributions, which were tracked during the MCTS search phase, differ significantly. To our knowledge, AUPO is the first abstraction algorithm to build abstractions based solely on reward distribution statistics.

The paper is structured as follows. First, in **Section 2**, we give an overview of domain knowledge independent abstraction tree searches. Next, in **Section 3** we formalize our problem setting, and lay the theoretical groundwork for understanding AUPO. This is followed by **Section 4** where we formalize AUPO and **Section 5** where we experimentally verify AUPO and discuss the experimental results. Lastly, in **Section 6** we summarise our findings and show avenues for future work.

## 2 RELATED WORK

The literature on abstraction-using planners is vast and ranges from abstractions for strategy games (Moraes & Lelis, 2018; Xu et al., 2023), card games such as Poker (Billings et al., 2003) to board games such as Go (Childs et al., 2008) or even hospital scheduling planners (Friha et al., 1997). Aside from such domain-specific abstractions, general abstraction methods are developed for continuous and/or partially observable domains (Hoerger et al., 2024) or learning-based abstractions such as learning and planning on abstract models (Ozair et al., 2021; Kwak et al., 2024; Chitnis et al., 2020), or building abstractions that rely on learned functions (e.g. a value function) (Fu et al., 2023). There are, however, only a handful of abstraction algorithms that are going to present next, that have the same scope as AUPO, which are non-learning-based, domain-independent action or state abstraction methods, for a discrete, fully-observable setting.

**State abstractions:** Jiang et al. (2014) were the first to propose a technique to automatically detect state abstractions in parallel to running a tree search. In regular intervals, they pause MCTS and group states within a layer when each action is pairwise approximately equivalent in the sense that their immediate rewards are similar and their transition probabilities to the node groups of the previous layer also lie within a threshold. To be able to detect any abstractions at all, they optimistically group all partially explored nodes within a layer and use a directed acyclic graph (DAG), allowing different state-action pairs to have the same successors, which is the basis for the abstraction buildup. Though the authors did not name this technique themselves, others refer to it as AS-UCT (Anand et al., 2015). The computed abstraction is only used in the tree policy by improving the UCB value where instead of an action's true visits and values, one instead inserts the sum of visits and values of all corresponding actions of nodes in the same abstract node.

AS-UCT can be improved by using different grouping conditions that allow for the detection of more symmetries, for example, by defining two states to be equivalent if their actions can be mapped to each other. This condition was first formulated by Ravindran & Barto (2004) and experimentally tested by Anand et al. (2015), who called this technique ASAM-UCT.

**State and action space abstractions:** One could detect even more symmetries by grouping states if, for each action in a node, there is at least one equivalent action in the other node and vice versa. Furthermore, one can abstract nodes and actions independently. Though they are primarily state abstractions, one can also implicitly view AS- and ASAM-UCT as action abstractions, however, two actions can only ever be abstracted if their parents are in the same abstract node. These ideas are combined in ASAP-UCT, which was also proposed by Anand et al. (2015). The successor of ASAP-UCT is called OGA-UCT (Anand et al., 2016) which improves the runtime and accuracy of ASAP-UCT by recomputing the abstraction only for frequently visited nodes thus ensuring the information contained in the abstraction does not lack behind the current search tree.

While AS-UCT builds the abstractions on an empirical model, ASAP-, ASAM-, and OGA-UCT rely on full knowledge of the problem's transition function which is not required by AUPO. While these methods apply the abstraction throughout the entire tree, AUPO only affects the root

node's actions during the decision policy. Hence, the usage of AUPO does not exclude the usage of OGA-UCT for example. Since both ASAM-UCT and AS-UCT are state abstractions they can only ever detect the equivalence of two sibling actions, a weakness that AUPO does not have. Additionally, all of the above-mentioned abstraction techniques require a DAG for search which could be impossible for a setting where either saving states or comparing them is infeasible or simply not possible. AUPO does not have this requirement.

**Refining abstractions:** All of the above-mentioned techniques can be thought of as pessimistic in that they only abstract actions or states when precise conditions are met. However, in environments where equivalences are the norm and not the exception, optimistic approaches can thrive. For example, PARSS by Hostetler et al. (2015) initially groups all successors of each state-action pair. As the search progresses, this coarse abstraction is refined by repeatedly splitting abstract nodes in half. Like PARSS, AUPO can also be viewed as a refining and optimistic abstraction algorithm, but whereas PARSS randomly refines its abstractions when it does not have access to additional state information, AUPO does so using statistical evidence. The method of fully abandoning an abstraction mid-search can also be seen as refining and has been coined Elastic MCTS by Xu et al. (2023). Though not fully domain-independent, another refining approach is given by Sokota et al. (2021), who group states based on a domain-specific distance function, and the maximal grouping distance shrinks as the search progresses.

## 3 FOUNDATIONS

We use finite Markov Decision Processes (MDP) (Sutton & Barto, 2018) as the model for sequential, perfect-information decision-making tasks. We use $\Delta(X)$ to denote the probability simplex of a finite, non-empty set $X$.

*Definition:* An *MDP* is a 6-tuple $(S, \mu_0, \mathbb{A}, \mathbb{P}, R, T)$ where the components are as follows:

- $S \neq \emptyset$ is the finite set of states, $T \subseteq S$ is the (possibly empty) set of terminal states, and $\mu_0 \in \Delta(S)$ is the probability distribution for the initial state.
- $\mathbb{A}\colon S \mapsto A$ maps each state $s$ to the available actions $\emptyset \neq \mathbb{A}(s) \subseteq A$ at state $s$ where $|A| < \infty$.
- $\mathbb{P}\colon S \times A \mapsto \Delta(S)$ is the stochastic transition function where we use $\mathbb{P}(s' \mid s, a)$ to denote the probability of transitioning from $s \in S$ to $s' \in S$ after taking action $a \in \mathbb{A}(s)$ in $s$.
- $R\colon S \times A \mapsto \mathcal{R}$ is the reward function that maps to the set of real-valued random variables.

Starting in $s_0 \sim \mu_0$, an MDP progresses from state $s_t$ to $s_{t+1}$ by first sampling an action $a_t \sim \pi(s_t)$ and then sampling $s_{t+1} \sim \mathbb{P}(\cdot|s_t, a_t)$ where $\pi$ is any agent for $M$. An agent $\pi\colon S \mapsto \Delta(A)$ for an MDP $M$ is a mapping from states to action distributions with $\pi(s)(a) = 0$ for any $a \notin \mathbb{A}(s)$. Crucially, an agent's output depends only on a single state. At each transition $M$ samples the reward $r_t \sim R(s_t, a_t)$.

In this paper, we consider only the finite horizon setting where the game ends after at most $h \in \mathbb{N}$ steps or earlier when a terminal state is reached. We call $h$ the horizon and any state-action-reward sequence $(s_0, a_0, r_0), \ldots, (s_n, a_n, r_n), s_{n+1}$ that can be reached a *trajectory*. If additionally $n + 1 = h$ or $s_{n+1} \in T$ we call this trajectory an *episode*.

The goal of any agent is to maximize its expected *return*. The return of an episode $\tau = (s_0, a_0, r_0), \ldots, (s_n, a_n, r_n), s_{n+1}$ is defined as the (possibly discounted) sum of rewards, i.e. $R(\tau) := \sum_{i=0}^{n} r_i \cdot \gamma^i$ where $0 < \gamma \leq 1$ is the *discount factor*. For any given state $s$, action $a \in \mathbb{A}(s)$, and maximum remaining steps $k \leq h$ we call $Q^*(s, a, k)$ the Q-value of $(s, a, k)$ and $V^*(s, k)$ the state value of $s$ (given $k$ remaining steps) which are defined as

$$Q^*(s, a, k) := \max_{\pi} \mathbb{E}_{\tau \sim \tau(\pi, s, a, k)}[R(\tau)], \tag{1}$$

$$V^*(s, k) := \max_{a \in \mathbb{A}(s)} Q^*(s, a, k) \tag{2}$$

where $\tau(\pi, s, a, k)$ denotes the trajectory distribution of an agent $\pi$ induced by starting at state $s$, directly applying $a$ and then playing according to $\pi$ for at most $k - 1$ steps or until a terminal state is reached. We write $Q^*(s, a) := Q^*(s, a, h)$ and $V^*(s) := V^*(s, h)$ and $V^* := \mathbb{E}_{s_0 \sim \mu_0}[V^*(s_0)]$.

Our AUPO method will heavily rely on and be compared to MCTS (for a detailed description, see Section A.9). The MCTS version we employ here, uses a greedy decision policy as well as the UCB tree policy.

# 4 METHOD

**The benefit of finding abstractions to the decision policy:** One component of MCTS is its decision policy which decides which action to take given the previously obtained search tree statistics. A common decision policy and the one we employ here is the greedy strategy where one picks the root node action with the highest Q-value (sum of all returns divided by visits).

The Q-value of each (root node) action-visits pair can be viewed as a real-valued random variable. Furthermore, these random variables are independent iff their corresponding actions are different. Let us assume that there are $n \in \mathbb{N}$ root actions in total. We denote the respective Q-value random variables by $Q_1, \ldots, Q_n$. For simplicity, let us assume that each root action has the same number of visits and that the optimal action is the same as $\arg \max_{1 \leq a \leq n} \mathbb{E}[Q_a]$.

Furthermore, let us assume that $\mathbb{E}[Q_1] = \cdots = \mathbb{E}[Q_k], k < n$. Though consequently the actions $a_1, \ldots, a_k$ are value-equivalent they suffer from an overestimation bias in the decision policy that worsens exponentially with increasing $k$. The decision policy is invariant under replacing $Q_1, \ldots, Q_k$ by the random variable $Q^m := \max(Q_1, \ldots, Q_k)$. Trivially, $\mathbb{E}[Q^m] \geq \mathbb{E}[Q_1]$ and more concretely, it holds that for any constant $c \in \mathbb{R}$

$$\mathbb{P}(Q^m \geq c) = 1 - \mathbb{P}(Q^m < c) = 1 - \prod_{i=1}^{k} \mathbb{P}(Q_i < c). \tag{3}$$

If we managed to detect that $\mathbb{E}[Q_1] = \cdots = \mathbb{E}[Q_k]$ and abstract them into a single random variable $\bar{Q} := \frac{Q_1 + \cdots + Q_k}{k}$, then not only can the previously mentioned overestimation bias be fully mitigated but we can even decrease the variance since

$$\text{Var}(\bar{Q}) = \frac{1}{k^2} \sum_{i=1}^{k} \text{Var}(Q_i). \tag{4}$$

**Finding abstractions by distribution comparisons:** The main idea of AUPO is to find and utilize action abstractions at the root node during the decision policy by comparing the reward distributions at depths $1, \ldots, D$ of the game tree. Initially, AUPO assumes all actions to be equivalent, however, if the reward distributions of two actions differ significantly at any depth, the two actions are separated.

**Building the abstraction** Let us assume we are in a state $s \in S$ with actions $a_1, \ldots, a_n$. After running standard MCTS for $m$ iterations, we have sampled $m$ trajectories where we denote the trajectories that started with action $a_j$ by $\tau_{i,j} = (a_{w_1}, r_1, s_1), (a_{w_2}, r_2, s_2), \ldots, (a_{w_{D_{i,j}}}, r_{D_{i,j}}, (s_{D_{i,j}})), a_{w_1} = a_j, 1 \leq i \leq m_j, m_1 + \cdots + m_n = m$. Consider the reward sequence $R_{d,j}$ obtained at depth $d$ after playing action $a_j$ at the root node i.e.

$$((R_{d,j})_i)_{1 \leq i \leq m_j} := r_d \text{ with } (a_{w_d}, r_d, s_d) = (\tau_{i,j})_d \tag{5}$$

where we define $r_d := 0$ in case $D_{i,j} < d < D$.

Though this is a heuristic assumption, we assume that all $R_{d,j}$ are samples from a stationary distribution $\mathcal{R}_{d,i}$ (this assumption would only hold if we performed a pure Monte Carlo search). Next, we compute the empirical mean and standard deviation for all $\mathcal{R}_{d,j}, d \leq D$ along with their confidence intervals for a fixed confidence level $q \in [0, 1]$. Any pair of actions $a_j, a_k$ has $2 \cdot D$ reward distributions associated with them which are $\mathcal{R}_{1,j}, \ldots, \mathcal{R}_{D,j}$ for $a_j$ and $\mathcal{R}_{1,k}, \ldots, \mathcal{R}_{D,k}$ for $a_k$. AUPO then groups $a_j, a_k$ if and only if all confidence intervals (both the mean and std intervals) up to depth $d \leq D$ of the pairs $(\mathcal{R}_{d,j}, \mathcal{R}_{d,i})$ overlap. If any confidence interval pair does not overlap, then $a_j, a_k$ are separated. Note that this induces a soft-abstraction where it is possible that for three actions $(a, b, c)$, $a$ is grouped with $b$, $b$ is grouped with $c$ but $a$ is not grouped with $c$.

Optionally, to ensure that in the limit, AUPO does not group non-value-equivalent actions, we may additionally separate two actions, if the distribution of their returns differs significantly (in the

sense that their mean and standard deviation confidence intervals do not overlap). The return of a trajectory is the (possibly discounted) sum of all its rewards. We call this option the return filter $\text{RF} \in \{0, 1\}$.

In theory, it would also be possible to do this distribution separation using either distance measures for probability distributions, such as the Wasserstein distance, or use statistical tests for determining whether two means or two standard deviations differ significantly. However, we found the mean and standard deviation to be sufficient descriptors for the underlying distributions whose computation only requires keeping track of the total number of samples, the sample sum, and the sum of squares, instead of saving every single reward at every depth. Furthermore, the confidence intervals only have to be computed once whereas we would have to compute test statistics and distribution distances for every single action pair, which can become a significant runtime overhead for large action spaces.

**Using the abstraction:** We use the abstractions during the decision policy only. AUPO transforms the decision policy into a two-step process. In the first step, we assign each action $a_j$ its abstract Q-value which is the sum of the returns divided by the sum of the visits of all actions $a_j$ is grouped with. We select the action $a^*$ that maximizes the abstract Q-value. Ties are broken randomly. In the second step, we select the action inside the abstraction of $a^*$ with the highest unabstracted/ground Q-value. This decision policy makes AUPO a generalization of the greedy decision policy as for both $q \in \{0, 1\}$ AUPO's decision policy degenerates to the greedy policy. While for $q = 0$ step two becomes redundant, for $q = 1$ step one becomes redundant. We summarize AUPO in the Appendix in Alg. 1.

**Theoretical guarantees:** The key innovation that makes AUPO work in practice (this will be shown empirically later) is that one does not only compare a single pair of distributions to differentiate a single action pair but rather one compares a number of distributions induced by that action pair. Using some simplifying assumptions, one can show that with an increase in $D$, the order of the number of samples required to differentiate two non-equivalent actions changes. More precisely, assume that AUPO is run on an MDP with $2D + 1$ states. The root state $s_0$ has two deterministic actions $a^{\text{down}}$ and $a^{\text{up}}$ that transition to $s_1^{\text{down}}$ and $s_1^{\text{up}}$ respectively which themselves have only a single deterministic action that transitions to $s_{i+1}^{\text{down}}$ or $s_{i+1}^{\text{up}}$ when $s_i^{\text{down}}$ or $s_i^{\text{up}}$ was the previous state. The rewards obtained at the two chains are Gaussian with means $\mu^{\text{down}} = (\mu_1^{\text{down}}, \ldots, \mu_D^{\text{down}})$ and $\mu^{\text{up}} = (\mu_1^{\text{up}}, \ldots, \mu_D^{\text{up}})$ and standard deviations $\sigma^{\text{down}} = (\sigma_1^{\text{down}}, \ldots, \sigma_D^{\text{down}})$ and $\sigma^{\text{up}} = (\sigma_1^{\text{up}}, \ldots, \sigma_D^{\text{up}})$. Furthermore, it is assumed that AUPO has access to the standard deviations when building the confidence intervals (which otherwise would be estimated by the empirical standard deviation). Now assume that both chains have been played $n$ times. The following statement (which is proven in the appendix Section A.1) can be made about AUPO's abstraction probability when neither the return-, nor std filter is used, the distribution tracking depth is equal to $D$, and confidence level $q \in (0, 1)$ is chosen:

$$\forall \varepsilon > 0 : \ \mathbb{P}[\text{AUPO abstracts } a^{\text{down}} \text{ and } a^{\text{up}}] \in \mathcal{O}(f(n)), f(n) = e^{-n \cdot (\varepsilon + \sum\limits_{k=1}^{D} w_i)} \tag{6}$$

where for $1 \leq i \leq D$: $w_i = \begin{cases} \frac{(\mu_i^{\text{down}} - \mu_i^{\text{up}})^2}{2(\sigma_i^{\text{down}} + \sigma_i^{\text{up}})^2}, & |\mu_i^{\text{down}} - \mu_i^{\text{up}}| \geq \frac{z^*}{\sqrt{n}}(\sigma_i^{\text{down}} + \sigma_i^{\text{up}}) \\ 1, & \text{otherwise} \end{cases}$, and $z^*$ is the critical value of the standard normal distribution for $q$ (e.g. $z^* \approx 1.96$ for $q = 0.95$).

**AUPO example:** Next, we illustrate on an instance of the IPPC problem SysAdmin how AUPO detects abstractions. A detailed explanation of this problem is given in the experiment Appendix A.7. Assume we are in a state where all computers, except one outer computer, are online. This is visualized in Fig. 1a. This state features exactly four value-equivalent action types. Idling, rebooting the offline computer (machine 3), rebooting (even though it is still online) the hub computer (machine 0), or rebooting any outer running computer (machines 1-2,5-9). Given enough trajectory samples, AUPO separates and subsequently detects these equivalences as follows.

*Idle action*: All actions except idling have the same immediate reward, the reboot cost. Therefore, the idle action is easily separated by considering only the mean of the 1-step reward distribution.

*Rebooting the offline computer*: This action can be separated from the others by the 2-step reward distribution, as it takes one step for the computer to be rebooted and then another step to receive the reward from the additional running computer. Though a little noisy, the 2-step reward will be on average 1 higher than that of the other actions. We quantitatively verified this in the Appendix in Tab. 2.

*Rebooting the hub computer*: This action can be separated from rebooting any of the outer running computers by the standard deviation of the 3-step reward. If we reboot the hub computer, we safeguard it from randomly crashing in the next step, which prevents the catastrophe where numerous other computers fail in the next step as they are connected to the then-broken hub computer. This scenario happens only rarely but when it does happen it is catastrophic, thus causing the 3-step reward of not rebooting the hub to have a relatively high variance compared to rebooting and thus protecting it. We quantitatively verified this in the Appendix in Tab. 1.

*Rebooting an outer running computer*: Since they are symmetric and thus have identical reward distributions at all downstream steps, AUPO optimistically assumes that are equivalent and thus abstract into a single action.

**Relation to other abstraction frameworks** In practice, AUPO is able to detect abstractions that ASAP could not because the latter requires the state graph to converge on states from which the abstraction building can be bootstrapped. Hence it is practically impossible for ASAP to detect equivalences that arise due to symmetry. For example, while it would be no problem for AUPO to detect that saving any of the four corner cells is equivalent in the Game of Life state visualized in Fig. 1b, ASAP would not be able to detect this with feasible computational resources. Game of Life is defined in the Appendix A.7.

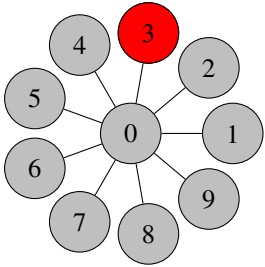
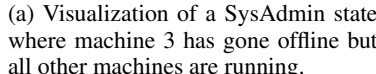
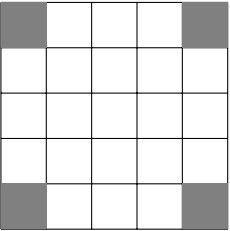

(a) Visualization of a SysAdmin state where machine 3 has gone offline but all other machines are running.

(b) A 5×5 Game of Life configuration with only the four corner cells alive.

Figure 1: Visualization of two environments considered in this paper.

Furthermore, ASAP struggles with a high stochastic branching factor. While AUPO is able to detect that rebooting any of the outer machines from the SysAdmin example in Section 4 is equivalent, ASAP is not able to detect these equivalences if two equivalent actions have not sampled the exact same set of successor from which there are $33554432 = 2^{25}$.

## 5 EXPERIMENTS

In this section, we will present the setup and results for the comparison of AUPO with MCTS showing that AUPO is the first and currently only tree-search abstraction algorithm that does neither require access to the transition probabilities, nor the model having deterministic rewards, nor requires a directed acyclic search graph but can outperform MCTS.

**Problem models:** The problem models that we tested AUPO on are either problems from the International Conference on Probabilistic Planning (IPPC) (Grzes et al., 2014) or appear throughout the literature. For the readers not familiar with these problem models, we give a high-level overview as well as a brief description of the environments' trivial value equivalences (which does not mean there aren't any other additional approximate equivalences) in the appendix in Section A.7. For details and the concrete instances, i.e. model parameter choices, we refer to

our publicly available implementation (Authors, 2025), which is the translation into C++ of the Relational Dynamic Influence Diagram (RDDL) (Sanner, 2011) descriptions of these environments found at the RDDL repository of Taitler et al. (2022). These environments were deliberately chosen as they appear throughout the abstraction literature (Anand et al., 2015; 2016; Hostetler et al., 2015; Yoon et al., 2008; Jiang et al., 2014) or have been used for planning competitions (Grzes et al., 2014), feature value-equivalent sibling actions, dense rewards, two theoretically necessary requirements for AUPO to yield any performance increase in the first place.

**Experiment setup and reproducibility:** For every experiment, we used a horizon of 50 episode steps. Since we are in the finite-horizon setting, we used a discount of $\gamma = 1$. We ran every experiment for at least 2000 episodes, and whenever we denote the mean return of this experiment we additionally provide a 99% confidence interval. We denote the confidence interval of any quantity by its mean and the half of the interval size, e.g. we would denote a return confidence interval $(1, 3)$ by $2 \pm 1$. For both MCTS and AUPO, we performed random playouts until the episode terminates. Additionally, as the problem models vary in their reward scale, we used a dynamic exploration factor that is given by $C \cdot \sigma$ where $\sigma$ is the empirical standard deviation of all Q values of the current search tree and $C \in \mathbb{R}^+$ is a parameter. For reproducibility, we released our implementation (Authors, 2025). Our code was compiled with g++ version 13.1.0 using the -O3 flag (i.e. aggressive optimization).

**Parameter-optimized performances:** First, we tested whether and in which environments AUPO can increase the parameter-optimized performance over MCTS. To do this, we considered the best AUPO performance when varying the parameters exploration constant $C \in \{0.5, 1, 2, 4, 8, 16\}$, distribution tracking depth $D \in \{1, 2, 3, 4\}$, using the return filter SF $\in \{0, 1\}$, using the return filter RF $\in \{0, 1\}$, and varying the confidence level $q \in \{0.8, 0.9, 0.95, 0.99\}$. Furthermore, since the standard UCB tree policy results in non-uniformly distributed visits, we also considered AUPO's performance when using a uniform root policy (denoted as U-AUPO) which has two main effects. Firstly, each action, even those that UCB would not exploit, receive visits, thus shrinking their confidence intervals, making them easier to separate from other actions. And secondly, we reduce the risk of separating reward distribution equivalent actions because in MCTS the distributions shift with an increasing visit count as MCTS starts to exploit.

We compare AUPO and U-AUPO to the performance of MCTS and MCTS with a uniform root policy U-MCTS, as well as RANDOM-ABS that is the same as AUPO except that for each action pair they are randomly abstracted at the decision policy with the probability $p_{\text{random}} \in \{0.1, 0.2, \ldots, 0.9\}$. Hence, RANDOM-ABS is equivalent to MCTS in the cases $p_{\text{random}} \in \{0, 1\}$. RANDOM-ABS verifies that the abstractions found by AUPO outperform randomly formed abstractions. Do reduce the amount of visuals; any RANDOM-ABS data points are simply the maximum of both RANDOM-ABS with a uniform root policy and standard root policy. The parameter-optimized performances in dependence of the iteration number are visualized in Fig. 2. The following key observations can be made:

**1)** AUPO can gain a clear performance **advantage** over MCTS (and RANDOM-ABS) in **11 out of the 14 here-considered environments**, in at least one iteration budget. In the environments, Academic Advising, Game of Life, Multi-armed bandit, Push Your Luck, Cooperative Recon, SysAdmin, and Traffic, AUPO maintains a clear performance edge for the majority of iteration budgets.

**2)** Expectedly, U-MCTS mostly performs worse than MCTS, however, the performance improvements between U-MCTS and U-AUPO is mostly significantly greater than the gap between MCTS and AUPO, showing the AUPO as suggested benefits from uniformly distributed visits. Notably, there is an environment, namely Cooperative Recon in which MCTS and U-MCTS perform evenly, where however, U-AUPO clearly outperforms AUPO. Also, in Saving both U-MCTS and U-AUPO outperform their non-uniform counterparts. Hence, using a uniform root policy can be a tool to improve the peak performance.

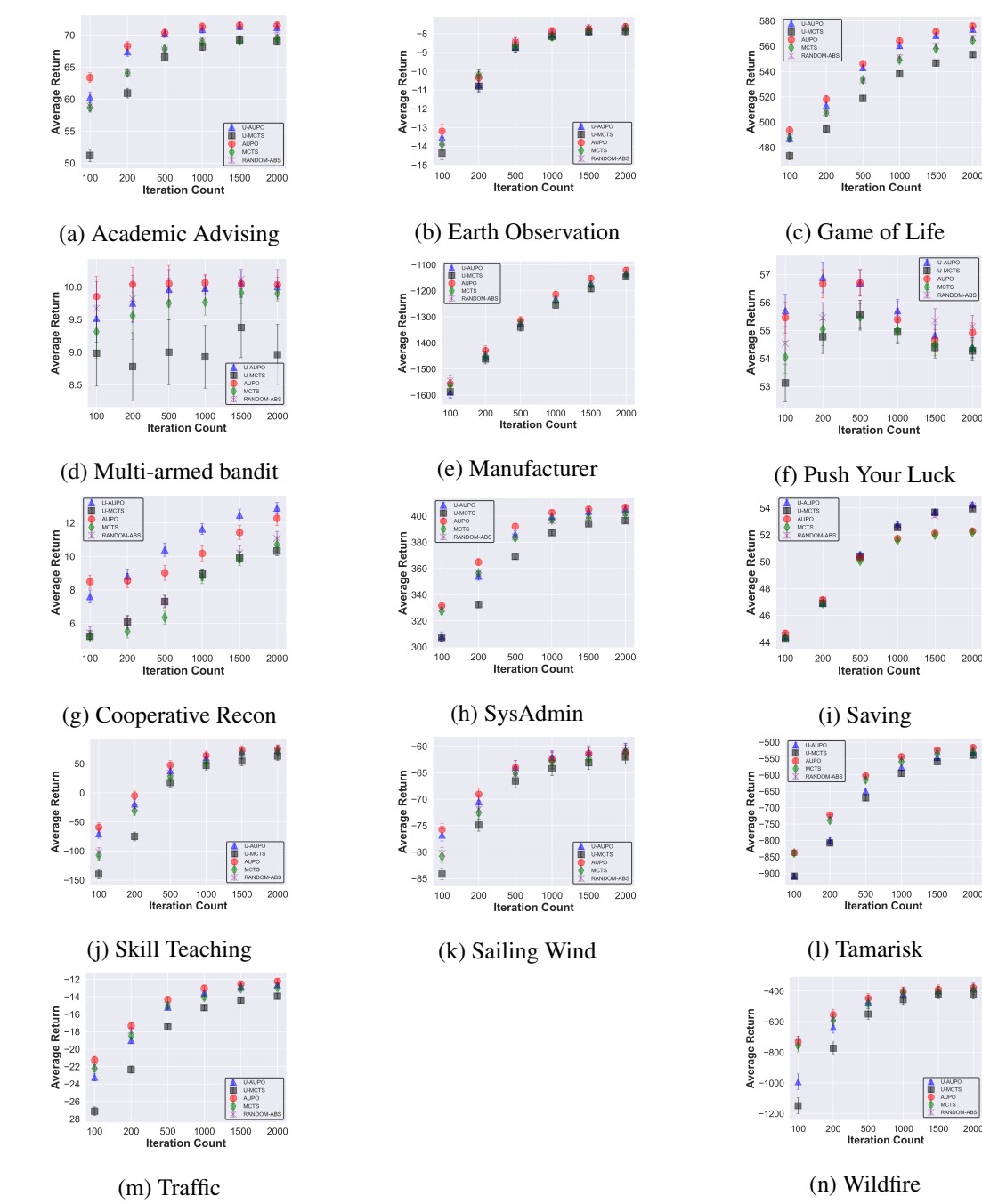

Figure 2: The performance graphs of in dependence of the MCTS iteration count of the parameter optimized versions of AUPO, MCTS, and RANDOM-ABS. The prefix U- denotes AUPO and MCTS using a uniform root policy.

**Generalization capabilities:** Next, we test AUPO's generalization capabilities. For this, we computed the pairings and relative improvement scores for all AUPO, U-AUPO, MCTS, U-MCTS, and RANDOM-ABS parameter combinations. These scores are Borda-like rankings of individual parameter-combinations and both lie in the interval $[-1, 1]$ (1 is the best value and -1 the worst) and they are formalized in the Appendix Section A.10. The results for all iteration budgets and environments combined are visualized in Fig. 3 and show that the best performances with respect to

both scores with large margins, are reached with AUPO. These results are qualitatively identical for each iteration budget which is presented in the Appendix Section A.11.

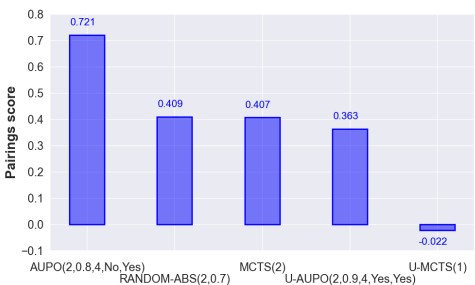
(a) Pairings score

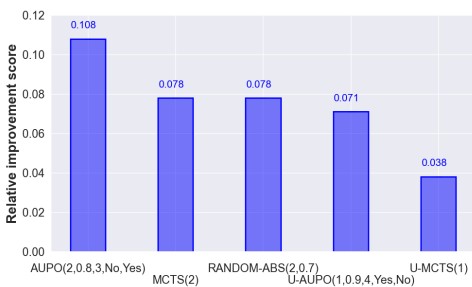
(b) Relative improvement score

Figure 3: The pairings and relative improvement scores across all environments and iteration budgets for different AUPO, U-AUPO (parameter format $(C, q, D, \text{RF}, \text{SF})$), MCTS, U-MCTS (parameter format $(C)$), and RANDOM-ABS (parameter format $(C, p_{\text{random}})$) agents. The bar charts show the top score reached by each agent type as well as the parameter combination to reach that score. In the case of RANDOM-ABS the score was reached with the standard root policy.

**Ablations:** Lastly, we are going to study the impact of the individual parameters. Instead of displaying the best-performing parameter set, we fix the parameter in question and max over the remaining parameters. With only a few exception such as Multi-armed bandit, the confidence level does only have a significant impact for low iteration counts if it has any impact all. In the low iteration count regime, lower confidences generally outperform higher confidence levels. Depending on the environment, the impact of the distribution tracking depth can be significant to non-existent. In cases where it does matter, high depths are always preferred with the only exception being Push Your Luck. Filters can be extremely beneficial to some environments, such as Multi-armed Bandit or Wildfire Whilst not causing any harm to the environment where it has little impact. We visualize the concrete performance values for this ablation in the Appendix Fig. 5 which shows the results when varying the confidence level, in Fig. 4, which shows the results when varying the distribution tracking depth, and in Fig. 6 that shows the results when varying either the std filter or the return filter.

## 6 LIMITATIONS AND FUTURE WORK

In this paper, we introduced a novel action abstraction algorithm that we call AUPO which only affects the decision policy of MCTS. We could experimentally show that AUPO outperforms MCTS in a wide range of environments that contain states with value-equivalent sibling actions. Though AUPO introduces four new parameters, their choice mostly has only a minor impact on performance.

First and foremost, for AUPO to achieve any performance gain, the environment must contain state-action pairs with the same parent that have similar $Q^*$ values, i.e. there need to be abstractions to be detected in the first place. Another key limitation of AUPO is that it is reliant on dense-rewards. For example, in binary-outcome zero-sum two-player games AUPO would have a hard time distinguishing actions, as only the return distribution can be used for differentiation. How this limitation can be overcome, is left as future work. Another weakness of AUPO is that it requires many visits for the distributions to be distinguishable; hence it cannot be used in low iteration settings and therefore not during the tree policy. Therefore, another area for future work is how to make AUPO much more sensitive to be able to deal with low iterations. Furthermore, for future work, as mentioned in the introduction, it could also be of interest to combine AUPO with other abstraction algorithms. For example, one may use state-of-the-art such as OGA-UCT (Anand et al., 2016) during the search phase, replacing only the decision policy with AUPO. In its current form, AUPO uses the same confidence level for each layer. However, it might be worth investigating if additional performance can be achieved by making this parameter layer-dependent.

# 7 REPRODUCIBILITY STATEMENT

In our experiment setup, we have a subsection called *Reproducibility* in which we provide a download link to the full codebase used for this project as well as compilation details. The codebase contains an elaborate README detailing the steps to reproduce the experiments.

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

# A  APPENDIX

## A.1  PROOF OF ABSTRACTION PROBABILITY THEOREM

In this section, Equation 6 from Section 4 is proven. Firstly, we will derive a general upper bound for the probability of confidence intervals overlapping and then use this result in the context of AUPO's abstraction mechanism.

**1)** Let $n \in \mathbb{N}$ and $X_1, \ldots, X_n, Y_1, \ldots, Y_n$ be i.i.d. Gaussian random variables with respective means and stds of $\mu_X \geq \mu_Y$ and $\sigma_X, \sigma_Y$. For any confidence level $q \in [0, 1]$, the confidence interval for $\mu_X$ (analogously $\mu_Y$) is of the form

$$[\overline{X} \pm \frac{z^*}{\sqrt{n}}] \tag{7}$$

where $z^* \in \mathbb{R}$ is the z-score for the given confidence level $q$ and $\overline{X} = \frac{1}{n}\sum_{k=1}^{n} X_i$ ($\overline{Y}$ is defined analogously). The probability that the confidence intervals for $\mu_X$ and $\mu_Y$ overlap is thus given by

$$\mathbb{P}[|\underbrace{\overline{X} - \overline{Y}}_{Z:=}| \leq \underbrace{\frac{z^*}{\sqrt{n}}(\sigma_X + \sigma_Y)}_{T:=}]. \tag{8}$$

Since $Z$ is Gaussian and the mean of $Z$ is $\mu_Z := \mu_X - \mu_Y$ and the std is $\sigma_Z := \frac{\sigma_X + \sigma_Y}{\sqrt{n}}$, and since $\mathbb{P}[|Z| \leq T] = \mathbb{P}[Z \leq T] - \mathbb{P}[Z \leq -T]$ one obtains

$$\mathbb{P}[|Z| \leq T] = \frac{1}{2}\left[\text{erf}\left(\frac{T + \mu_Z}{\sqrt{2}\sigma_Z}\right) + \text{erf}\left(\frac{T - \mu_Z}{\sqrt{2}\sigma_Z}\right)\right] \tag{9}$$

using the identity $\Phi(\frac{x-\mu}{\sigma}) = \frac{1}{2}(1 + \text{erf}(\frac{x-\mu}{\sigma\sqrt{2}}))$ that holds for any Gaussian with mean $\mu$ and std $\sigma$ where $\Phi$ is the CDF for the standard Gaussian distribution and erf is the Gauss error function. Next,

using that erf is an odd function with range $(-1, 1)$, yields

$$\mathbb{P}[|Z| \leq T] = \frac{1}{2}\left[-\operatorname{erfc}\left(\frac{\mu_Z + T}{\sqrt{2}\sigma_Z}\right) + \operatorname{erfc}\left(\frac{\mu_Z - T}{\sqrt{2}\sigma_Z}\right)\right] \leq \frac{1}{2}\operatorname{erfc}\left(\frac{\mu_Z - T}{\sqrt{2}\sigma_Z}\right) \text{ with } \operatorname{erfc} := 1 - \operatorname{erf}. \tag{10}$$

Next, two cases are differentiated. If $\mu_Z - T < 0$, we simply bound $\mathbb{P}[|Z| \leq T]$ by 1. In the other case, $\mu_Z - T \geq 0$, one can use an upper bound derived by Giuseppe Abreu (Abreu, 2012) to further estimate this expression in terms of the exponential function. Concretely this yields,

$$\frac{1}{2}\operatorname{erfc}\left(\frac{\mu_Z - T}{\sqrt{2}\sigma_Z}\right) \leq \frac{1}{50}e^{-x^2} + \frac{1}{2(x+1)}e^{-x^2/2} \leq e^{-x^2/2}, x = \frac{\mu_Z - T}{\sigma_Z}, \tag{11}$$

which is a function of the form

$$e^{-\tilde{\lambda}_1 + \tilde{\lambda}_2\sqrt{n} - w \cdot n}, \text{ with } w = \frac{(\mu_X - \mu_Y)^2}{2(\sigma_X + \sigma_Y)^2}; \tilde{\lambda}_1, \tilde{\lambda}_2 \in \mathbb{R}^+. \tag{12}$$

**2)** By definition, AUPO using no return or std filter with a distribution tracking depth $D$ only abstracts $a^{\text{down}}$ and $a^{\text{up}}$ iff their mean confidence intervals up to depth $D$ all overlap. Since in this two-chain MDP, all reward distributions are independent, the probability of all confidence intervals overlapping, is given as the product of the individual ones overlapping, we can use the previously obtained results about a single pair of confidence intervals to obtain the following for every $\varepsilon > 0$

$$\mathbb{P}[\text{AUPO abstracts } a^{\text{down}} \text{ and } a^{\text{up}}] \leq e^{-\lambda_1 + \sqrt{n} \cdot \lambda_2 - n \cdot \sum_{k=1}^{D} w_i} \in \mathcal{O}(f(n)), \lambda_1, \lambda_2 \in \mathbb{R}^+, \tag{13}$$

where $f(n) = e^{-n \cdot (\varepsilon + \sum_{k=1}^{D} w_i)}$ and for $1 \leq i \leq D$:

$$w_i = \begin{cases} \frac{(\mu_i^{\text{down}} - \mu_i^{\text{up}})^2}{2(\sigma_i^{\text{down}} + \sigma_i^{\text{up}})^2}, & |\mu_i^{\text{down}} - \mu_i^{\text{up}}| \geq \frac{z^*}{\sqrt{n}}(\sigma_i^{\text{down}} + \sigma_i^{\text{up}}) \\ 1, & \text{otherwise} \end{cases} \tag{14}$$

.

This proves the original statement. $\qquad\square$

## A.2 AUPO PSEUDOCODE

---

**Algorithm 1:** AUPO

---

**Parameters:** $q$ , $D$, $filter\_std$, $filter\_return$, $mcts\_args$

**Input:** $state$

    // Run MCTS and collect reward distribution data

**1** $n = \text{num\_actions}(state)$ , $R[d, j] = [] \; \forall d, j$

**2 for** $i = 1 \dots mcts\_iterations$ **do**

**3**      sample MCTS trajectory with rewards $r_1, \dots, r_{D^*}$ and first action $a_j$

**4**      **for** $d = 1 \dots D$ **do**

**5**          $R[d, j]$.append($r_d$ **if** $d \le D^*$ **else** 0)

**6**      **end**

**7**      $R^*[j]$.append$\big(r_1 + \dots + r_{\min(D, D*)}\big)$

**8 end**

    // Compute confidence intervals

**9 for** $j = 1 \dots n$ **do**

**10**      **for** $d = 1 \dots D$ **do**

**11**          $mean\_interval[d, j] = \text{mean\_conf\_interval}(R[d, j], q)$

**12**          $std\_interval[d, j] = \text{std\_conf\_interval}(R[d, j], q)$

**13**      **end**

**14**      $return\_mean\_interval[j] = \text{mean\_conf\_interval}(R^*[j], q)$

**15**      $return\_std\_interval[j] = \text{std\_conf\_interval}(R^*[j], q)$

**16 end**

    // Compute abstractions

**17 for** $i = 1 \dots n$ **do**

**18**      $abstract\_visits = 0, abstract\_value = 0$

**19**      $abstraction[i] = \{\}$

**20**      **for** $j = 1 \dots n$ **do**

**21**          $abstracted = $ **true**

**22**          **for** $d = 1 \dots D$ **do**

**23**              **if** $mean\_interval[d, j] \cap mean\_interval[d, i] == \emptyset$ *or* $filter\_std$ *and*
             $std\_interval[d, j] \cap std\_interval[d, i] == \emptyset$ **then**

**24**                  $abstracted = $ **false**

**25**              **end**

**26**          **end**

**27**          **if** $filter\_return$ *and (*
         $return\_mean\_interval[d, j] \cap return\_mean\_interval[d, i] == \emptyset$ *or* $filter\_std$ *and*
         $return\_std\_interval[d, j] \cap return\_std\_interval[d, i] == \emptyset$) **then**

**28**              $abstracted = $ **false**

**29**          **end**

**30**          **if** $abstracted$ **then**

**31**              $abstract\_visits + = action\_visits(j)$

**32**              $abstract\_value + = action\_returns(j)$

**33**              $abstraction[i]$.insert($j$)

**34**          **end**

**35**          $abstract\_Q[i] = \frac{abstract\_value}{abstract\_visits}$

**36**      **end**

**37 end**

    // Action selection

**38** $abs\_action = \arg \max\limits_{i=1 \dots n} abstract\_Q[i]$

**39** $ground\_action = \arg \max\limits_{i \in abstraction[abs\_action]} Q[i]$

**40 return** $ground\_action$;

---

## A.3 ABLATION: DISTRIBUTION TRACKING DEPTH $D$

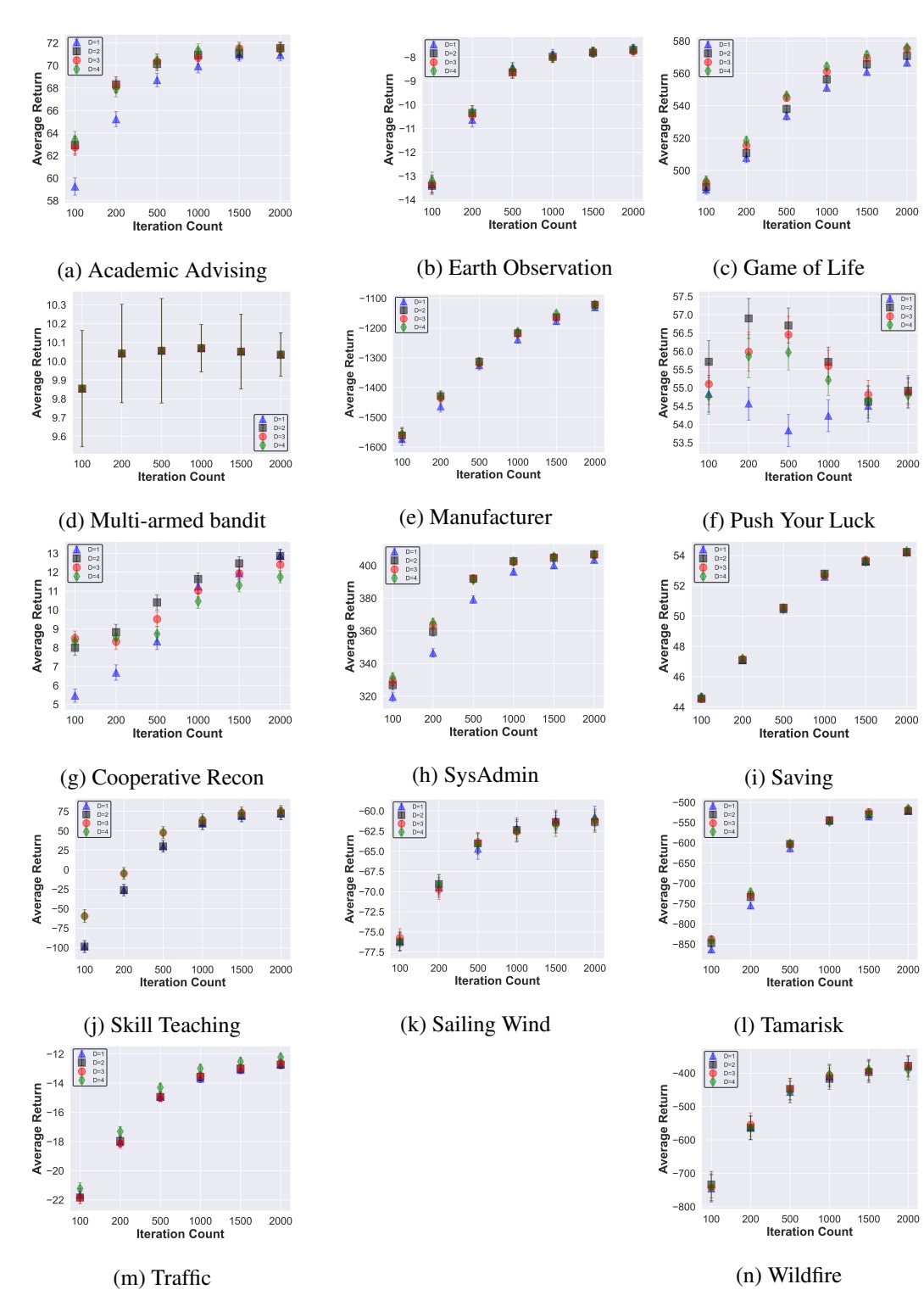

Figure 4: The performance graphs of in dependence of the MCTS iteration count of the parameter optimized versions of AUPO using different fixed values for the distribution tracking depth $D$.

## A.4 PERFORMANCES IN DEPENDENCE OF THE CONFIDENCE LEVEL $q$

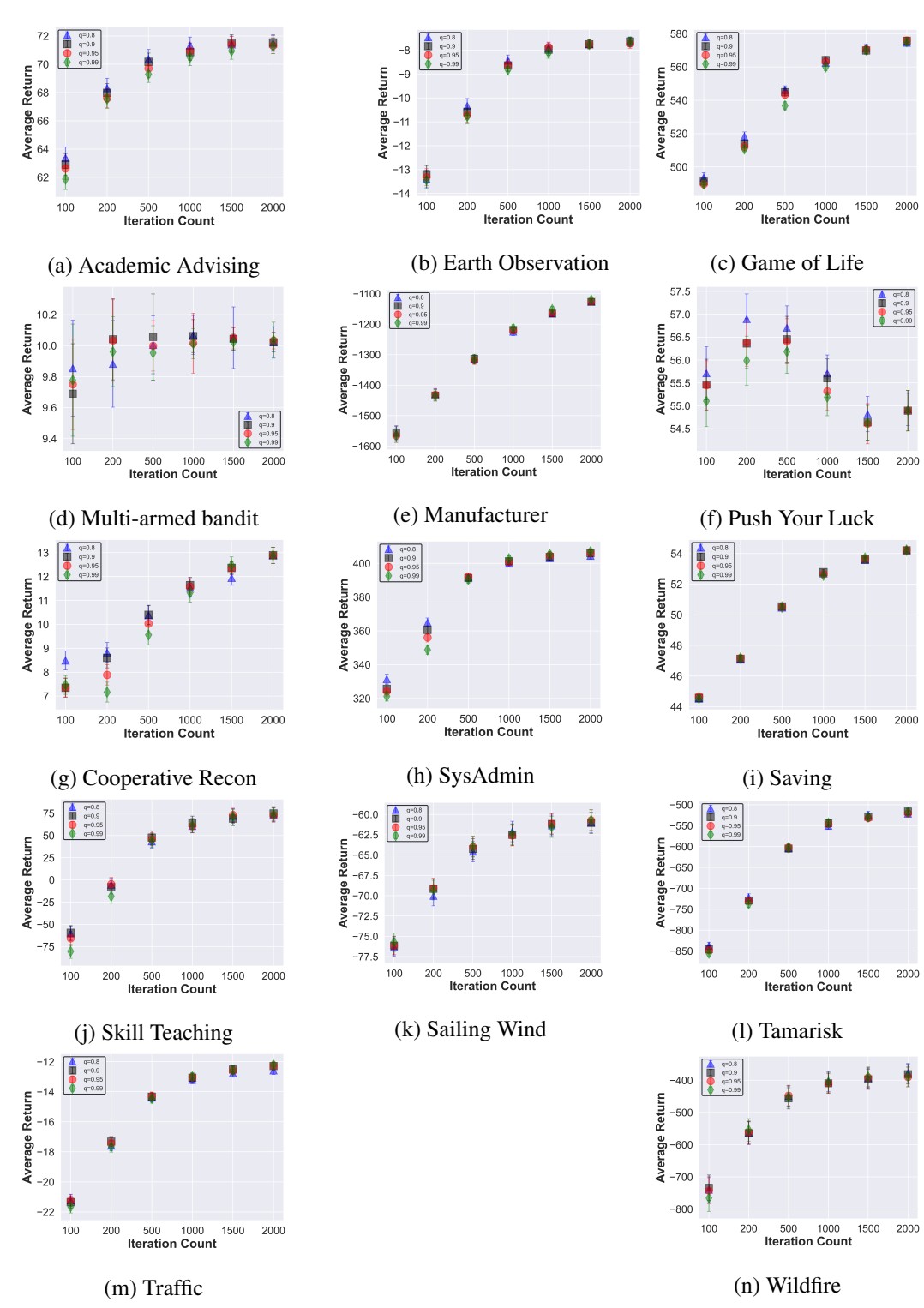

Figure 5: The performance graphs of in dependence of the MCTS iteration count of the parameter optimized versions of AUPO using different fixed values for the confidence $q$.

## A.5 PERFORMANCES WHEN USING DIFFERENT FILTER COMBINATIONS

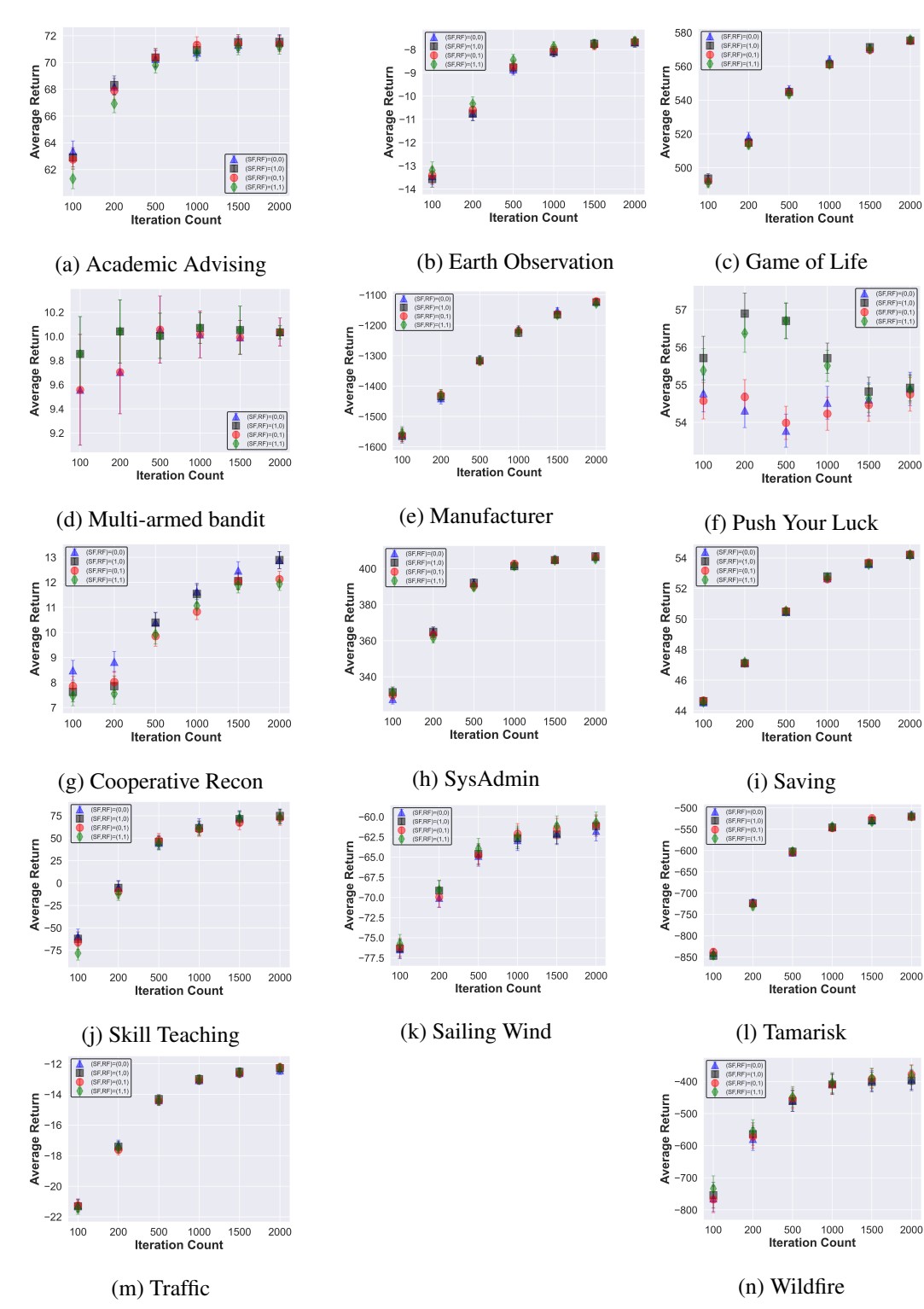

Figure 6: The performance graphs of in dependence on the MCTS iteration count of the parameter optimized versions of AUPO using different fixed filter settings. Both the return (RF) and std filter (SF) are varied.

## A.6 Reward distribution confidence intervals for SysAdmin

Table 1: 3-step standard deviation 95% confidence intervals for the reward distribution after a different number of MCTS iterations on the SysAdmin state of Fig. 1a. Note that with higher iteration counts, rebooting the hub can be separated from the remaining actions.

| Iterations | Hub (0) | 1 | 2 | 3 | 4 | 5 | 6 | 7 | 8 | 9 | Idle |
|---|---|---|---|---|---|---|---|---|---|---|---|
| 1000 | (0.94, 1.26) | (0.88, 1.18) | (1.20, 1.60) | (1.15, 1.54) | (1.06, 1.43) | (0.83, 1.12) | (0.87, 1.16) | (0.92, 1.24) | (0.99, 1.33) | (1.14, 1.52) | (1.02, 1.37) |
| 2000 | (0.85, 1.04) | (1.06, 1.31) | (1.13, 1.39) | (1.01, 1.24) | (1.13, 1.39) | (1.45, 1.78) | (0.94, 1.15) | (1.11, 1.37) | (1.33, 1.64) | (1.20, 1.48) | (1.19, 1.47) |
| 3000 | (0.85, 1.01) | (1.21, 1.43) | (1.05, 1.25) | (1.15, 1.36) | (1.13, 1.33) | (1.25, 1.47) | (0.99, 1.18) | (1.03, 1.22) | (1.19, 1.40) | (1.20, 1.42) | (1.22, 1.44) |
| 4000 | (0.85, 0.99) | (1.21, 1.40) | (1.16, 1.34) | (1.07, 1.24) | (1.14, 1.32) | (1.18, 1.37) | (1.07, 1.24) | (1.10, 1.28) | (1.10, 1.28) | (1.17, 1.35) | (1.16, 1.35) |

Table 2: 2-step mean 95% confidence intervals for the reward distribution after different numbers of MCTS iterations on the SysAdmin state of Fig. 1a. Note that even with very low iteration counts, rebooting machine 3 can easily be separated from the other actions.

| Iterations | Hub (0) | 1 | 2 | 3 | 4 | 5 | 6 | 7 | 8 | 9 | Idle |
|---|---|---|---|---|---|---|---|---|---|---|---|
| 250 | (7.75, 8.20) | (7.83, 8.32) | (7.86, 8.26) | (8.60, 9.30) | (7.61, 8.22) | (7.36, 8.07) | (7.74, 8.30) | (7.66, 8.18) | (7.25, 8.00) | (7.40, 8.22) | (7.60, 8.32) |
| 500 | (7.88, 8.22) | (7.74, 8.15) | (7.57, 8.11) | (8.69, 9.06) | (7.63, 8.01) | (7.59, 8.01) | (7.62, 8.09) | (7.58, 8.05) | (7.74, 8.11) | (7.77, 8.16) | (7.48, 7.92) |
| 750 | (7.96, 8.25) | (7.70, 8.05) | (7.72, 8.07) | (8.66, 9.00) | (7.61, 7.97) | (7.80, 8.12) | (7.85, 8.17) | (7.73, 8.06) | (7.62, 8.03) | (7.67, 8.03) | (7.61, 8.00) |
| 1000 | (7.90, 8.19) | (7.82, 8.10) | (7.69, 8.01) | (8.51, 8.83) | (7.74, 8.02) | (7.96, 8.22) | (7.80, 8.07) | (7.83, 8.13) | (7.79, 8.08) | (7.90, 8.19) | (7.84, 8.15) |

## A.7 Problem models

In the following, we provide a brief description of each domain/environment that was used in this paper. Some of these environments can be parametrized (e.g., choosing a concrete map size for *Sailing Wind*). The concrete parameter settings can be found in the *ExperimentConfigs* folder in our publicly available GitHub repository (Authors, 2025). In the following, for the reader's convenience, we re-introduced the relevant environment descriptions from the survey paper (Schmöcker & Dockhorn, 2025) as well as added new ones for those not contained in the survey. For a detailed description of these environments, we refer to our implementation.

**Academic Advising**: The Academic Advising domain was introduced by Guerin et al. (2012) and modified for the IPPC 2014 (Grzes et al., 2014) (the version used here). The agent is a student whose goal is to pass certain academic classes. Formally, the state is an element in $\{P, \text{NP}, \text{NT}\}^n$ (representing for each course whether it has been passed, not been passed, or not taken), and the agent's action is to choose a course to take. The course outcome depends on the states of the prerequisite courses. The episode terminates if all courses have been passed. The agent incurs costs for taking and redoing courses. In the original IPPC version, the agent would also always receive a negative reward as long as there is one mandatory course that has not been passed. We increased the reward density, but let this negative reward be dependent on the number of missing mandatory courses. Furthermore, we also added a reward for every course passed.

Aside from symmetries that arise from the course dependency graph, all actions where one retakes a course that has already been passed are trivially equivalent.

**Cooperative Recon**: This domain models a robot tasked with discovering signs of life on a foreign planet. The robot operates on a two-dimensional grid populated with various objects of interest and a central base. When the robot reaches an object of interest, it can perform surveys to detect the presence of water and, subsequently, life. The probability of detecting life increases if water is first identified. If life is successfully detected, the robot can then photograph the object - this is the only action that yields a reward. Each use of a detector carries a risk of failure, which may render the detector unusable or reduce its reliability. Detectors can be repaired, but only at the base location.

**Earth Observation**: This problem, proposed by Hertle et al. (2014), models a satellite orbiting Earth while performing photographic observations. Each state corresponds to a position on a two-dimensional grid, where the satellite's longitudinal location and the latitude at which its camera is aimed are represented. Additionally, certain designated cells have associated weather levels that influence observation quality. Weather conditions change stochastically at each time step, independent of the agent's actions. The agent can choose to idle, take a photograph of the current target cell, or adjust the camera's focus by incrementing or decrementing the $y$-position (latitude). A reward is granted when a designated cell is photographed, with the reward magnitude depending

on the prevailing weather in that cell.

**Game of Life**: The original game of life by John Conway (Gardner, 1970) is a cellular automaton and was modified into a stochastic MDP as a test problem for the International Probabilistic Planning Competition (IPPC) (Sanner & Yoon, 2011) by introducing noise to the deterministic state transition, setting the current number of alive cells as the reward, and allowing the agent to choose one cell per round that will survive to the next round with a high probability. States are elements in $\{0, 1\}^{n \times n}$ describing whether there is an alive cell at each cell on a grid. In the original problem, one could not only save cells but revive dead cells, however, this action space would have been too big to obtain meaningful reward distributions given our iteration numbers.

Besides symmetries, all actions where one would save a cell that would survive due to the deterministic rules anyway, are trivially equivalent.

**Manufacturer**: In this domain, the agent is responsible for managing a manufacturing company with the objective of selling goods to customers. To do so, the agent must first produce the goods, which may involve constructing factories and procuring the necessary input materials. A key challenge lies in the stochastic fluctuations of goods' market prices.

**Multi-armed bandit**: Multi-armed bandits (MAB) (Kuleshov & Precup, 2014) are 1-step MDPs. Each action $1 \leq a \leq n$ is called an arm, and its execution yields an immediate random reward sampled from the probability distribution associated with the $a$-th arm. We use Gaussians as the reward distributions.

All actions whose associated arms have the same mean are equivalent. We deliberately chose a MAB instance with a high number of equivalences.

**Push Your Luck**: In Push Your Luck the agent has to decide which of $n$, $m$-sided, not-necessarily fair dice or cash-out. If cashed-out, the agent receives a reward dependent on all dice faces that are marked. Faces are marked if they have been rolled (each face is shared by all $n$ dice). However, if the agent rolls an already marked face, or rolls two unmarked faces at the same time, all markings are removed.

**Sailing Wind**: Originally proposed by Robert Vanderbei (Vanderbei, 1996), the goal of Sailing Wind is to move a ship that starts at $(1, 1)$ on an $n \times n$ grid to $(n, n)$ with minimal cost. There is no consistent use of a transition and reward function throughout the literature. There may just be two available actions (*down*, *right*) (Jiang et al., 2014) or up to seven (each adjacent cell except the one facing a stochastic wind direction) (Anand et al., 2015). The reward at each step is $(-1 + W)$ where $0 \leq W \leq 4$ is dependent on the current wind direction which stochastically changes its direction at each step independent of the player's actions.

**Saving**: Saving is introduced by Hostetler et al. (2015), where the agent aims to maximize accumulated wealth over time. At each step, the agent can choose one of three actions: Invest, Borrow, or Save. Borrow provides an immediate reward of 2 but imposes a penalty of -3 after $n$ time steps. Once this action is taken, it cannot be repeated until the delayed penalty is applied. Save yields an immediate reward of 1 with no further consequences. Invest offers no immediate reward but enables the agent to take the Sell action within the next $m$ time steps. The agent cannot invest again until either the Sell action is executed or $m$ steps have elapsed. If Sell is chosen, then the agent receives a reward equal to the current price level that changes stochastically and independently of the agent's actions.

**Skill Teaching** In Skill Teaching, the agent takes the role of a tutor that is tasked with increasing the proficiency level of a student at various skills. The student can have one of three proficiency levels at each skill: Low, medium, and high. The skills from a prerequisite graph, giving the student higher chances of learning a new skill the higher the prerequisites' levels of proficiency. Difficulty arises from the proficiency levels decaying if the corresponding skill wasn't practised. This decay is deterministic for skills at medium proficiency and stochastic for those at high proficiency.

**Tamarisk**: The Tamarisk domain models the spread of an invasive plant in a river system.

This problem was also used for the IPPC 2014 (Grzes et al., 2014) and inspired by the work of Muneepeerakul et al. (2007). The river is split into $n$ reaches, each containing $k$ possible slots for plants. Each slot can either be empty, occupied by a native plant, by the invasive Tamarisk plant or by both. Both Tamarisk and native plants can randomly spawn at empty slots, however, Tamarisk can stochastically spread to neighboring reaches with a higher probability of spreading downstream. The agent's actions are to restore native plants, eradicate Tamarisk, or to idle. All non-idling actions target an entire reach at once where they randomly but independently of each other succeed at each slot. The goal is to balance minimizing the Tamarisk spread with the high action costs.

**Traffic**: In this environment, the agent is tasked with simultaneously controlling a number of traffic lights with the goal of minimizing traffic jams. This traffic is modelled as a directed graph, however, some edges are only available depending on the state of a traffic light. Each vertex may either contain a car or not.

**SysAdmin**: Proposed by Guestrin et al. (2003), a SysAdmin instance is a graph (describing a network topology) with $n \in \mathbb{N}$ vertices. The state space is $\{0, 1\}^n$ (describing which machines are currently operating) and the action space is $\{1, \ldots, n\} \cup \{\text{IDLE}\}$ (describing which machine to reboot or whether to idle). At each step, an agent receives a reward equal to the number of working machines as well as a punishment if a machine has been rebooted. A reboot deterministically ensures that the rebooted machine is working again in the next step, however, this action has no effects beyond this step. Machines can randomly fail at each step; however, this probability increases with the number of failed neighbors.

Action equivalences depend on the topology that is being used. For the one displayed in Fig. 1a which we also use for the experiments, rebooting any of the outer computers with the same state is equivalent.

**Wildfire**: Also used for the IPPC 2014 (Grzes et al., 2014) and based of the work by Karafylldis and Thanailakis (Karafylldis & Thanailakis, 1997), Wildfire models the spread of a fire on a grid. Each grid cell is either untouched, burning, or out-of-fuel meaning that no new fire can ignite at this cell. If a cell is untouched it can at each time step randomly ignite with the probability increasing exponentially in the number of neighboring burning cells. The neighborhood is defined on an instance level with most instances choosing the 8-neighborhood and manually cutting a handful of neighborhood connections between individual cells. The agent is punished for each burning cell and additionally punished for predefined target cells that are burning. The agent's actions are to idle, to cut out the fuel of a cell, or to put out a fire at any cell. These actions always succeed, with putting out a fire incurring the highest costs.

Trivial equivalences here are to cut out fuel where fuel has already been cut out or to put out a fire where there is no fire.

## A.8 RUNTIME MEASUREMENTS

We validate the claim that AUPO adds only a minor runtime overhead over vanilla MCTS for high iteration budgets, the following table, Tab. 3 lists the average decision-making times for each environment of AUPO compared to MCTS for 100 and 2000 iterations on states sampled from a distribution induced by random walks. This shows that while AUPO adds a significant overhead for low iteration budgets, the impact of the decision policy and therefore AUPO's runtime overhead vanishes. Note, though, that this runtime is both heavily implementation and hardware-dependent, and more efficient implementations might reduce this overhead. In particular, we are using highly optimized environment implementations that could be the runtime bottleneck in more complex environments.

Table 3: Average decision-making times of AUPO and MCTS in milliseconds for 100 and 2000 iterations. For AUPO the most computational heavy version has been used, which uses $p = 0.8, D = 4$, the return- and std filter. This data was obtained using an Intel(R) Core(TM) i5-9600K CPU @ 3.70GHz. The data shows a median runtime overhead of $\approx 8\%$ for 100 iterations and $\approx 4\%$ for 2000 iterations.

| Domain | AUPO-100 | MCTS-100 | AUPO-2000 | MCTS-2000 |
|---|---|---|---|---|
| Academic Advising | 1.15 | 1.59 | 23.71 | 25.36 |
| Cooperative Recon | 2.41 | 2.57 | 52.25 | 54.21 |
| Earth Observation | 7.03 | 6.95 | 130.57 | 136.73 |
| Game of Life | 3.93 | 4.31 | 65.72 | 65.18 |
| Manufacturer | 10.31 | 9.82 | 185.76 | 186.81 |
| Sailing Wind | 1.90 | 2.01 | 34.34 | 35.15 |
| Saving | 0.82 | 0.87 | 17.41 | 18.24 |
| Skills Teaching | 2.42 | 2.61 | 52.20 | 53.91 |
| SysAdmin | 1.20 | 1.58 | 22.97 | 24.29 |
| Tamarisk | 2.78 | 2.88 | 47.84 | 48.95 |
| Traffic | 3.05 | 3.85 | 61.23 | 63.74 |
| Triangle Tireworld | 1.25 | 1.32 | 25.43 | 27.22 |
| Push Your Luck | 2.26 | 2.47 | 43.21 | 45.38 |
| Multi-armed bandit | 0.16 | 1.16 | 3.13 | 4.33 |
| Wildfire | 1.48 | 2.20 | 34.22 | 35.73 |

## A.9 MONTE CARLO TREE SEARCH

AUPO heavily relies on Monte Carlo Tree Search (MCTS) which we are going to describe now. Let $M$ be a finite horizon MDP. On a high level, MCTS repeatedly samples trajectories starting at some state $s_0 \in S$ where a decision has to be made until a stopping criterion is met. The final decision is then chosen as the action at $s_0$ with the highest average return. In contrast to a pure Monte Carlo search, MCTS improves subsequent trajectories by building a tree from a subset of the states encountered in the last iterations which is then exploited. In contrast to pure Monte Carlo search, MCTS is guaranteed to converge to the optimal action.

An MCTS search tree is made of two components. Firstly, the state nodes, that represent states and Q nodes that represent state action pairs. Each state node, saves only its children which are a set of Q nodes. Q nodes save both its children which are state nodes and the number of and the sum of the returns of all trajectories that were sampled starting at the Q node.

Initially, the MCTS search tree consists only of a single state node representing $s_0$. Until some stopping criterion is met, the following steps are repeated.

1. **Selection phase**: Starting at the root node, MCTS first selects a Q node according to the so-called *tree policy*, which may use the nodes' statistics, and then samples one of the Q node's successor states. If either a terminal state node, a state node with at least one non-visited action (partially expanded), or a new Q node successor state is sampled, the selection phase ends.

   A commonly used tree policy (**and the one we used**) that is synonymously used with MCTS is Upper Confidence Trees (UCT) (Kocsis & Szepesvári, 2006) which selects an action that maximizes the Upper Confidence Bound (UCB) value. Let $s \in S$ and $V_a, N_a$ with $a \in \mathbb{N}$ be the return sum and visits and of the Q nodes of the node representing $s$. The UCB value of any action $a$ is then given by

$$\text{UCB}(a) = \underbrace{\frac{V_a}{N_a}}_{\text{Q term}} + \underbrace{\lambda \sqrt{\frac{\log\left(\sum_{a' \in \mathbb{A}(s)} N_{a'}\right)}{N_a}}}_{\text{Exploration term}}. \tag{15}$$

   The exploration term quantifies how much the Q term could be improved if this Q node was fully exploited and is controlled by the exploration constant $\lambda \in \mathbb{R} \cup \{\infty\}$. If one chose

$\lambda = 0$, the UCT selection policy becomes the greedy policy and for $\lambda = \infty$, the selection policy becomes a uniform policy over the visits. In case of equality, some tiebreak rule has to be selected, which is typically a random tiebreak. From here, will use MCTS and UCT (MCTS with UCB selection formula) synonymously.

2. **Expansion**: Unless the selection phases ended in a terminal state node, the search tree is expanded by a single node. In case the selection phase ended in a partially expanded state node, then one unexpanded action is selected (e.g. randomly, or according to some rule), the corresponding Q node is created and added as a child and one successor state of that Q node is sampled and added as a child to the new Q node. If the selection phase ended because a new successor of a Q node was sampled, then a state node representing this new state is added as a child to that Q node.

3. **Rollout/Simulation phase**: Starting at the state $s_{rollout}$ of the newly added state node of the expansion phase (or at a terminal state node reached by the selection phase), actions according to the *rollout policy* are repeatedly selected and applied to $s_{rollout}$ until a terminal state is reached. All states encountered during this phase are not added to the search tree.

4. **Backpropagation**: In this phase, the statistics of all Q nodes that were part of the last sampled trajectory that corresponds to a path in the search tree are updated by incrementing their visit count and adding the trajectory's return (of the trajectory starting at the respective Q node) to their return sum statistic.

Once the MCTS search tree has been built (by reaching an iteration limit in our case) and statistics have been gathered, the final decision is made by the *decision policy* that in our MCTS version simply chooses the action with the highest final Q value.

## A.10 DEFINITION OF RELATIVE IMPROVEMENT AND PAIRINGS SCORE

In the main experimental section, we evaluated AUPO with respect to the relative improvement and pairings score, which are formalized here. While the pairings score is calculated by summing over the number of tasks where some agent performed better than another, the relative improvement score also takes the percentage of the improvement into account; however, it is prone to outliers. Hence, we considered both scores to paint the full picture.

Concretely, let $\{\pi_1, \ldots, \pi_n\}$ be $n$ agents (e.g., concrete parameter settings for possibly different base algorithms such as AUPO or MCTS) where each agent was evaluated on $m$ tasks (in this paper, a task will always be a given MCTS iteration budget and an environment) where $p_{i,k} \in \mathbb{R}$ denotes the performance of agent $\pi_i$ on the $k$-th task.

**Definition:** The *pairings score matrix* $M \in \mathbb{R}^{n \times n}$ is defined as

$$M_{i,j} = \frac{1}{m-1} \sum_{1 \leq k \leq m} \mathrm{sgn}(p_{i,k} - p_{j,k}) \tag{16}$$

where sgn is the signum function. The *pairings score* $s_i \leq i \leq n$ is given by

$$s_i = \frac{1}{n-1} \sum_{1 \leq l \leq n, l \neq i} M_{i,l}. \tag{17}$$

**Definition** The *relative improvement matrix* $M \in \mathbb{R}^{n \times n}$ is defined as

$$M_{i,j} = \frac{1}{m-1} \sum_{1 \leq k \leq m} \frac{p_{i,k} - p_{j,k}}{\max(|p_{i,j}|, |p_{j,k}|)} \tag{18}$$

and the *relative improvement score* $s_i \leq i \leq n$ is given by

$$s_i = \frac{1}{n-1} \sum_{1 \leq l \leq n, l \neq i} M_{i,l}. \tag{19}$$

## A.11 Pairings and Relative Improvement Scores

Table 4: The pairings and relative improvement scores for the **100, 200, and 500 iterations** setting for the parameters combination of AUPO, U-AUPO, RANDOM-ABS, MCTS, and U-MCTS with the highest respective scores as well as the concrete parameters used to reach that score. The parameters and environments used to obtain these scores are the same as the experiments of Section 5. The parameter format for AUPO and U-AUPO is $(C, q, D, RF, SF)$, the format RANDOM-ABS is $(C, p_{\text{random}})$, and for both MCTS and U-MCTS is $(C)$. For RANDOM-ABS the best scores are obtained using the standard root policy.

100 iterations relative improvement score.

| Parameters | Score |
|---|---|
| AUPO(2,0.8,4,No,No) | 0.120 |
| U-AUPO(16,0.8,4,No,No) | 0.068 |
| RANDOM-ABS(2,0.9) | 0.055 |
| MCTS(2) | 0.049 |
| U-MCTS(4) | $-0.006$ |

100 iterations pairings score.

| Parameters | Score |
|---|---|
| AUPO(2,0.8,3,No,Yes) | 0.742 |
| RANDOM-ABS(2,0.7) | 0.360 |
| U-AUPO(1,0.8,4,Yes,No) | 0.319 |
| MCTS(2) | 0.301 |
| U-MCTS(0.5) | $-0.099$ |

200 iterations relative improvement score.

| Parameters | Score |
|---|---|
| AUPO(2,0.8,3,No,Yes) | 0.137 |
| RANDOM-ABS(1,0.8) | 0.073 |
| U-AUPO(0.5,0.9,4,Yes,No) | 0.068 |
| MCTS(2) | 0.062 |
| U-MCTS(1) | $-0.005$ |

200 iterations pairings score.

| Parameters | Score |
|---|---|
| AUPO(2,0.8,3,Yes,Yes) | 0.826 |
| RANDOM-ABS(2,0.9) | 0.438 |
| MCTS(2) | 0.368 |
| U-AUPO(0.5,0.8,4,Yes,No) | 0.330 |
| U-MCTS(0.5) | $-0.020$ |

500 iterations relative improvement score.

| Parameters | Score |
|---|---|
| AUPO(2,0.9,4,Yes,No) | 0.139 |
| RANDOM-ABS(2,0.4) | 0.109 |
| U-AUPO(0.5,0.9,3,Yes,No) | 0.107 |
| MCTS(2) | 0.105 |
| U-MCTS(0.5) | 0.069 |

500 iterations pairings score.

| Parameters | Score |
|---|---|
| AUPO(2,0.9,4,Yes,Yes) | 0.793 |
| U-AUPO(2,0.9,4,Yes,Yes) | 0.459 |
| MCTS(2) | 0.431 |
| RANDOM-ABS(1,0.7) | 0.417 |
| U-MCTS(0.5) | $-0.007$ |

Table 5: The pairings and relative improvement score for the **1000, 1500, and 2000 iterations** setting for the parameters combination of AUPO, U-AUPO, RANDOM-ABS, MCTS, and U-MCTS with the highest respective score as well as the concrete parameters used to reach that score. The parameters and environments used to obtain these scores are the same as the experiments of Section 5. The parameter format for AUPO and U-AUPO is $(C, q, D, RF, SF)$, the format RANDOM-ABS is $(C, p_{\text{random}})$, and for both MCTS and U-MCTS is $(C)$. For RANDOM-ABS the best scores are obtained using the standard root policy.

1000 iterations relative improvement score.

| Parameters | Score |
| --- | --- |
| AUPO(2,0.9,2,Yes,Yes) | 0.108 |
| RANDOM-ABS(2,0.9) | 0.089 |
| U-AUPO(1,0.9,4,Yes,Yes) | 0.089 |
| MCTS(2) | 0.084 |
| U-MCTS(1) | 0.057 |

1000 iterations pairings score.

| Parameters | Score |
| --- | --- |
| AUPO(2,0.9,4,Yes,Yes) | 0.770 |
| U-AUPO(1,0.9,4,Yes,Yes) | 0.499 |
| RANDOM-ABS(2,0.9) | 0.447 |
| MCTS(2) | 0.417 |
| U-MCTS(1) | 0.036 |

1500 iterations relative improvement score.

| Parameters | Score |
| --- | --- |
| AUPO(2,0.99,2,Yes,Yes) | 0.099 |
| MCTS(2) | 0.084 |
| RANDOM-ABS(2,0.8) | 0.083 |
| U-AUPO(1,0.8,4,Yes,Yes) | 0.083 |
| U-MCTS(1) | 0.061 |

1500 iterations pairings score.

| Parameters | Score |
| --- | --- |
| AUPO(2,0.9,4,Yes,Yes) | 0.763 |
| U-AUPO(2,0.9,4,Yes,Yes) | 0.532 |
| MCTS(2) | 0.438 |
| RANDOM-ABS(2,0.7) | 0.418 |
| U-MCTS(1) | 0.026 |

2000 iterations relative improvement score.

| Parameters | Score |
| --- | --- |
| AUPO(2,0.95,4,Yes,Yes) | 0.098 |
| RANDOM-ABS(2,0.7) | 0.087 |
| MCTS(2) | 0.086 |
| U-AUPO(1,0.99,4,Yes,Yes) | 0.086 |
| U-MCTS(1) | 0.059 |

2000 iterations pairings score.

| Parameters | Score |
| --- | --- |
| AUPO(2,0.95,4,Yes,Yes) | 0.753 |
| U-AUPO(2,0.95,4,Yes,Yes) | 0.538 |
| RANDOM-ABS(2,0.8) | 0.532 |
| MCTS(2) | 0.487 |
| U-MCTS(1) | 0.028 |

