# OpenReview forum: "AUPO - Abstracted until proven otherwise: A reward distribution based abstraction algorithm"
_ICLR.cc/2026/Conference — ICLR 2026 Conference Withdrawn Submission_

### Official Review · Reviewer_5vV5 · 2025-10-27

**Soundness:** 3
**Presentation:** 3
**Contribution:** 3
**Rating:** 6
**Confidence:** 4

**Summary:**

The paper describes an abstraction mechanism based on the reward distributions generated by different actions at the root of MCTS search. Different actions whose reward distributions at different depths are not statistically distinguishable are combined into a single node. A 2 level action selection first selects the abstract node that scores the highest, and then selects an action based on the standard MCTS scoring from the selected abstract node. Empirical results show that AUPO outperforms well on most of the benchmark planning domains.

**Strengths:**

The paper presents a general and simple approach that seems to work well across a range of domains.
The paper is well written and the results are thoroughly evaluated. AUPO seems to outperform MCTS in most domains.

**Weaknesses:**

The title "Abstraction until proven otherwise" is not very informative. Consider a more informative name for the method and the title, perhaps "reward-distribution guided abstarctions" or something like that.

There is no comparison to any abstraction methods although several have been discussed, e.g., AS-UCT, OGA-UCT etc. Admittedly some of these methods have stronger requirements such as knowing the transition model and may be inapplicable in some cases. While the comparison may not be totally fair, it is still useful to see what the relative improvements are.

The authors make the point that their method only works at the root level and hence can be combined with other abstraction techniques that work on the trees. It would be good to see if such combinations are practical or introduce additional complexities.

The method is only applicable for dense reward settings.

Minor comments:
L 035. "costly retrained" -> "expensively retrained"
L 048. Fix "exist that ....exist"
L 104. "lack behind" -> "lag behind"
L 143. "agent" is used instead of policy, possibly because it is stochastic. Agent is an overused term. I prefer policy or stochastic policy.
L 212. "R_{d,i} must be R_{d,k}"
L 213. "soft abstractions" How do you deal with them since each action should go into exactly one cluster?
L 267."(machines 1-2,5-9)" -> "(machines 1-2,4-9)"?
L 338. "C \sigma" A little more explanation of what role it plays in MCTS will be useful here.
L 345. "using the return filter SF" is repeated twice.
L 357. Stray comment: "Do reduce the amount of visuals"
L 430. Please give one line explanations of pairings score and relative improvement scores in the text.

**Questions:**

Can you compare the approach to one or two other abstraction techniques?

---

### Official Review · Reviewer_uAv6 · 2025-11-01

**Soundness:** 2
**Presentation:** 2
**Contribution:** 2
**Rating:** 2
**Confidence:** 3

**Summary:**

This paper introduces AUPO, an action abstraction algorithm which can be added to the decision policy of MCTS. The core principle of AUPO is to group actions at the root node where the layerwise statistics of the reward distributions at all layers are similar. The algorithm then selects actions by choosing the highest-value abstract action, then the highest-ground-value within that abstract action. Experiments are conducted on 14 simple environments from IPPC and the planning literature, which show that AUPO commonly outperforms the baselines of normal MCTS and AUPO with random abstractions.

**Strengths:**

The algorithm avoids issues with comparable baselines like needing full transition probabilities or DAGs. It does appear to produce performance increases, although I have some concerns here (see weaknesses). I also like that the algorithm is conceptually simple.

**Weaknesses:**

- I'm concerned about the evaluation practices used to compare AUPO to the baselines. In particular, around "we considered the best AUPO performance when varying the parameters…", it sounds like AUPO was optimized over 5+ hyperparameters, and we're taking the maximum, but the baselines may not have had the same treatment. How do I know that the performance gains of AUPO aren't just coming from more tuning?
- The settings are a bit toy, and I'd be curious to know if AUPO is effective in more complex applications of MCTS, e.g. board games. Perhaps AUPO is not applicable here due to the sparse reward condition.
- The quality of the writing is a weaker point, some sections are quite dense/verbose.
- "These environments were deliberately chosen as they … feature value-equivalent sibling actions, dense rewards, two theoretically necessary requirements for AUPO to yield any performance increase in the first place." This makes me concerned that AUPO is not very general. Can the authors say more about what kinds of environments have value-equivalent siblings? I'd like to understand how restrictive this is.
- The zero padding used when trajectories terminate early is questionable to me – I worry this would skew the reward distributions a lot / prevent good abstractions from being formed.

**Questions:**

- "Note that this induces a soft-abstraction where it is possible that for three actions (a, b, c), a is grouped with b, b is grouped with c but a is not grouped with c." How is action selection handled in this case?
- I am concerned that the random baseline is too easy of a comparison, but I might be missing something. Can the authors say more about why they did not compare to at least one of the baselines from the literature, such as ASAP?

---

### Official Review · Reviewer_MVsv · 2025-11-03

**Soundness:** 1
**Presentation:** 2
**Contribution:** 1
**Rating:** 0
**Confidence:** 4

**Summary:**

Monte Carlo tree search uses state/action abstraction to reduce search complexity in order to improve the depth of search along promising paths of the search tree. This paper introduces an abstraction algorithm to be used during decision time after search to select the optimal action.

**Strengths:**

Originality: The algorithm they introduce appears to be using a novel technique.

**Weaknesses:**

This paper lacks any significant contribution. The main contribution the authors tout is an abstraction method to be used during decision time after MCTS has been performed. I do not really understand the motivation for this --- the main reason abstractions are used during search is that the search space can be very large. Abstractions allow one to reduce the search space by aggregating similar states and actions into abstracted states and actions. This reduces the branching factor at the each layer of tree affording one the ability to build a deeper tree for more accurate outcome estimates and share information between similar states and actions. This paper introduces an abstraction algorithm for use *after* search is completed, which makes no sense. When making decisions, one desires more refinement after search, not less.

Even considering this to be used as a selection policy after tree search, the procedure makes no sense. It effectively performs an extra, more-computationally expensive iteration of search giving it unfair advantage over other selection policies. This makes comparisons against other selection policies unfair.

They also compared their selection policy against a single baseline selection policy: selecting the action with the highest value. I take that to mean they select the action with the highest estimated average. This is only one such selection process among many and one that is not reliably a good selection policy either. It is more common to select the action with the highest sample count or the highest lower confidence value. These policies are less vulnerable to the effects of variance in the outcomes.

I have a few gripes with their evaluation:
- The authors state the report the results of the parameter settings for AUPO that were the best. That is, it reads as if they ran the algorithm with varying parameter values and reported the results for this parameter sweep. Firstly, did they do that provide this courtesy to MCTS?
- Secondly, this is improper practice. Once the parameter sweep is done, you have run the algorithm on a separate test set using the best parameters.

Additionally, the paper is not well-written. There are several lines that make no sense and their are a number of terms and notation used without being defined or explained.

**Questions:**

- Line 85: "... their transition probabilities to the node groups of the previous layer also lie within a threshold." : Surely, the probabilities *from* the previous layer, right?

- Line 178: "Though consequently the actions $a_1, \dots, a_k$ are value-equivalent they suffer from an overestimation bias in the decision policy that worsens exponentially with increasing $k$." : Are you missing a comma?

- You have to define or explain $\mathbb{E}$ and $\mathbb{P}$.

- Trajectories are defined in two different ways.

- Line 266: rebooting should also include computer 4

**Details Of Ethics Concerns:**

This submissions contains parts that are identical to parts in other submissions.

---

> ### Author Response · Authors · 2025-11-18
> **Addressing ethical concerns of plagiarism and dual submission**
>
> We submitted 4 papers to this conference. Those do not represent a dual-submission but independent algorithmic developments based on a unified benchmark. Each of our submitted papers are in the general area of abstraction-based search. However, each submission studies a different algorithmic approach and makes its own primary technical and empirical contributions warranting a unique submission. These algorithms, their theoretical analyses, and their empirical results are non-overlapping. Each paper can be read and evaluated independently.
>
> To the best of our understanding, the conference allows multiple submissions from the same group as long as they present distinct, non-redundant scientific contributions, which is exactly how we structured these papers.
>
> Taking the reviewers concerns seriously, we still decided to withdraw our submission to make sure that even such minor similar sounding passages will be restructured before resubmission to this or another conference.

---

### Note · Authors · 2025-11-18

I have read and agree with the venue's withdrawal policy on behalf of myself and my co-authors.